# The Long-Term Change of Latent Heat Flux over the Western Tibetan Plateau

**Na Li [1,2], Ping Zhao [1,2,*], Jingfeng Wang [3] and Yi Deng [4]**

[1] State Key Laboratory of Severe Weather, Chinese Academy of Meteorological Sciences, Beijing 100081, China; nina_cams@163.com

[2] Collaborative Innovation Center on Forecast and Evaluation of Meteorological Disasters, Nanjing University of Information Science and Technology, Nanjing 210044, China

[3] School of Civil and Environmental Engineering, Georgia Institute of Technology, Atlanta, GA 30332, USA; jingfeng.wang@ce.gatech.edu

[4] School of Earth and Atmospheric Sciences, Georgia Institute of Technology, Atlanta, GA 30332, USA; yi.deng@eas.gatech.edu

[*] Correspondence: zhaop@cma.gov.cn; Tel.: +86-010-6840-8149

**Abstract:** The Tibetan Plateau (TP) has been experiencing warming and wetting since the 1980s. Under such circumstances, we estimated the summer latent heat flux (*LE*) using the maximum entropy production model driven by the net radiation, surface temperature, and soil moisture of three reanalysis datasets (ERA5, JRA-55, and MERRA-2) at the Ali site over the western TP during 1980–2018. Compared with the observed *LE* of the Third Tibetan Plateau Atmospheric Scientific Experiment, the coefficient of determination, root-mean-square error, and mean bias error of the estimated summer *LE* are 0.57, 9.3 W m$^{-2}$, and −2.25 W m$^{-2}$ during 2014–2016, respectively, which are better than those of *LE* of the reanalysis datasets. The estimated long-term summer *LE* presents a decreasing (an increasing) trend of −7.4 (1.8) W m$^{-2}$ decade$^{-1}$ during 1980–1991 (1992–2018). The *LE* variation is closely associated with the local soil moisture influenced by precipitation, glacier, and near-surface water conditions at the Ali site. The summer soil moisture also presents a decreasing (an increasing) trend of −0.082 (0.022) decade$^{-1}$ during 1980–1991 (1992–2018). The normalized difference vegetation index generally shows the consistent trend with *LE* at the Ali site.

**Keywords:** latent heat flux; western Tibetan Plateau; MEP model; long-term variation

---

## 1. Introduction

The Tibetan Plateau (TP), with an average elevation of 4500 m, is the highest plateau in the world. It plays an important role in elevating the heat sources into the middle troposphere over the Eurasian continent in summer, and profoundly influences the Asian summer monsoon, regional energy and water cycles, and environment changes [1]. The TP has been experiencing warming and wetting since the mid-1950s, which has led to an array of complex meteorological, hydrological, and ecological changes [2–6]. Therefore, to better understand the land-atmosphere interactions underlying regional and global climate pattern, the reliable estimations of surface thermal effects of the TP are essential. One component of the surface energy balance is the surface latent heat flux (*LE*). It represents the heat flux from the Earth's surface to the atmosphere associated with the surface evaporation or water [7,8]. Several intensive field experiments have been conducted off and on to measure *LE* over the TP since the 1970s [9–12]. However, the field experiments sites on the TP cannot cover the entire TP region; in particular, the sites are mainly distributed over the central-eastern TP, sparse in the western TP.

Considering the limited spatio-temporal coverage of the field experiments observations, the high spatio-temporal *LE* over the TP have been estimated using observational station data collected by the

China Meteorological Administration (CMA, Beijing), satellite remote sensing data, and reanalysis data [13,14]. The long-term routine meteorological data provided by CMA (i.e., wind speed, air temperature, humidity, and so on) have been used to estimate *LE* via the bulk transfer method in previous studies [15,16]. However, these results are insufficiently representative due to the relatively sparse CMA observation networks in the TP. Besides, it is also a challenge to estimate the moisture transfer coefficient. Because of the high spatio-temporal resolution of remote sensing and reanalysis datasets, some studies applied them to investigate the energy budget and the climatology and variability of *LE* over the TP [17,18]. Han et al. (2017) concluded that *LE* over the entire TP increased from 2001 to 2012. Song et al. (2017) presented that the actual evaporation estimated with meteorological observation and satellite remote sensing data decreased during 2000–2010. Even so, their study periods are too short to reveal long-term variations of *LE* well. The results reported in Han et al. (2017) and Song et al. (2017) are also contradictory to each other. In addition, climatic and environmental factors affecting the long-term change of *LE* in the western TP are still unclear.

Besides the *LE* estimation methods described above, a new approach, the maximum entropy production (MEP) model, has been developed to estimate the surface heat fluxes on bare soil, vegetation, water/snow/ice, and ocean surfaces [19]. The MEP model has been successfully applied to estimate the surface heat fluxes over the alpine steppe terrain of the central TP [20]. However, the MEP model has not yet been applied to estimate *LE* in the bare soil condition over the western TP.

Therefore, the objective of this study is to reveal the climatological variations of *LE* at the Ali site over the western TP in the warming and wetting climate of TP during 1980–2018 and the main factors associated with these variations. The applicability of the MEP model in estimating *LE* at the Ali site is validated based on the intensive field experimental data of the Third Tibetan Plateau Atmospheric Scientific Experiment (TIPEX-III) covering summers during 2014–2016.

## 2. Materials and Methods

### 2.1. Materials

The Ali site selected for this study is located at the western TP (32.49° N, 80.10° E) and covered with bare soil, with an elevation of 4255 m (Figure 1a). To validate the MEP model, we downloaded the TIPEX-III observed sensible heat flux (*SH*) and *LE* data at the Ali site from August 2014 to January 2017 from the website (http://data.cma.cn/tipex). They were measured by eddy covariance and hereafter referred to as $SH_{EC}$ and $LE_{EC}$, respectively. Following Li et al. (2019), the quality of the half-hourly $SH_{EC}$ and $LE_{EC}$ at the Ali site was controlled. We processed the half-hourly flux observations into daily values. The monthly $SH_{EC}$ and $LE_{EC}$ for the period from August 2014 to January 2017 are presented in Table 1. It is seen that the values of $LE_{EC}$ in summer (June through September) are greater than those in the other seasons. Therefore, we focused on summer in this study. The energy balance ratio, that is, $EBR = (SH + LE)/(R_n - G)$, was evaluated to ensure the usability of observations for the study site and period, in which $R_n$ is the net radiation and $G$ is the ground heat flux. In the absence of the observed $G$, it is estimated using the method of Li et al. (2019). Our result shows that the mean value of *EBR* is 0.90 in summers during 2014–2016, thereby indicating the reliability of the TIPEX-III summer surface heat fluxes at the Ali site. Moreover, the daily observed air temperature ($T_a$) and precipitation at the Ali site in summers from 1980 to 2018 were also used in this study.

The summer daily (monthly) $R_n$, skin temperature ($T_s$), soil moisture ($S_m$) in the top layer, and *LE* provided by the six reanalysis datasets were used in this study during 2014–2016 (1980–2018). The six datasets are the ECMWF ERA5 reanalysis (hereafter ERA5; [21]), the ECMWF interim reanalysis (hereafter ERA-Interim; [22]), the Japanese 55-Year Reanalysis (hereafter JRA-55; [23]), the Modern-Era Retrospective Analysis for Research and Application version 2 (hereafter MERRA-2; [24]), the NCEP/NCAR Reanalysis 1 (hereafter NCEP-I; [25]), and the NCEP-DOE Reanalysis 2 (hereafter NCEP-II; [26]). These reanalysis datasets were also applied to the MEP-based estimations on a daily/monthly scale during 2014–2016/1980–2018. We used the normalized difference vegetation index

(NDVI) of Advanced Very High Resolution Radiometer-Based Global Inventory Modelling and Mapping Studies (GIMMS3g) during 1981–2015 and the Moderate Resolution Imaging Spectroradiometer/Terra Vegetation Indices (MODIS) 16-Day L3 Global 250 m SIN Grid V006 from 2000 to 2018. These NDVI datasets have been widely used to indicate the variation of vegetation [27]. The bilinear interpolation method was adopted to interpolate the regular grid data to the appropriate point at the Ali site.

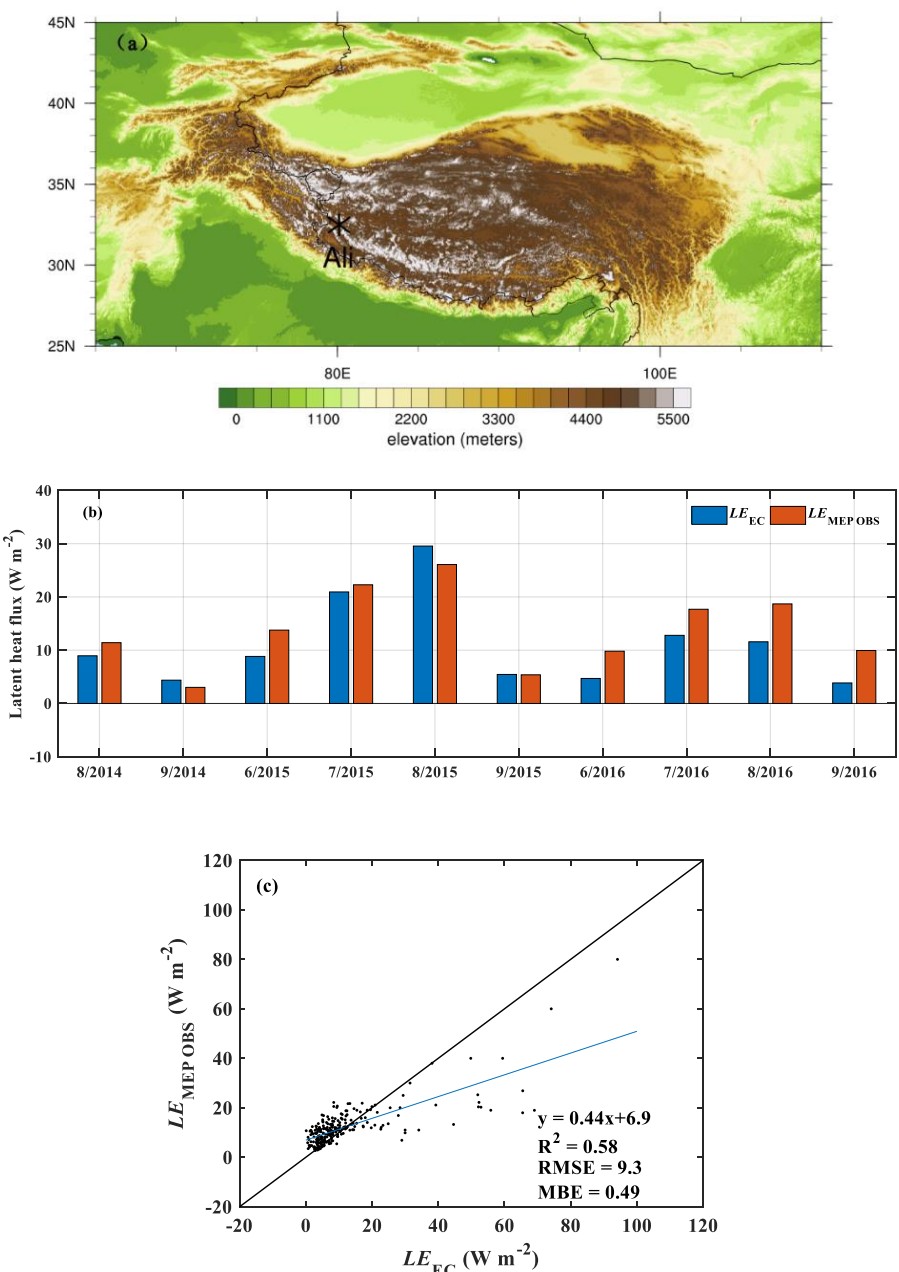

**Figure 1.** (**a**) The Ali location over the Tibetan Plateau (TP). Colors show the topographic elevation (m); (**b**) the monthly means of the observed latent heat flux ($LE_{EC}$) and the latent heat flux ($LE_{MEP\ OBS}$) modeled by the maximum entropy production (MEP) model and observations at the Ali site; (**c**) the comparison between $LE_{EC}$ and $LE_{MEP\ OBS}$ at the Ali site in summers from 2014 to 2016. In Figure 1c, the number of samples is 245, the blue line represents slope, and x and y represent $LE_{EC}$ and $LE_{MEP\ OBS}$, respectively.

**Table 1.** The monthly averages of the observed sensible heat flux ($SH_{EC}$) and latent heat flux ($LE_{EC}$) at the Ali site from August 2014 to January 2017.

|  | Jan | Feb | Mar | Apr | May | Jun | Jul | Aug | Sep | Oct | Nov | Dec |
|---|---|---|---|---|---|---|---|---|---|---|---|---|
| $SH_{EC}$ (W/m$^2$) | 22.0 | 31.7 | 45.6 | 62.4 | 69.5 | 64.8 | 52.6 | 53.4 | 52.1 | 39.3 | 24.1 | 16.1 |
| $LE_{EC}$ (W/m$^2$) | 0.22 | 0.57 | 1.37 | 0.54 | 1.78 | 6.77 | 16.86 | 16.69 | 4.56 | 1.51 | 0.24 | 0.04 |

*2.2. Methods*

In this study, the MEP model was used to calculate *LE*. The equations are as follows.

$$LE = R_n - \frac{LE}{B(\mu)}\left(\frac{B(\mu)}{\mu}\frac{I_s}{I_0}\left|\frac{LE}{B(\mu)}\right|^{-\frac{1}{6}} + 1\right),$$ (1)

$$B(\mu) = 6\left(\sqrt{1 + \frac{11}{36}\mu} - 1\right), \quad \mu(T_s, q_s) = \frac{L_v{}^2 q_s}{C_p R_V T_s^2}.$$ (2)

The nonlinear algebraic Equation (1) has a unique solution of *LE* through an implicit iterative method. $I_s$ ($= \sqrt{\rho C \lambda}$) is the thermal inertia of the surface layer material (J m$^{-2}$K$^{-1}$s$^{-1/2}$), $\rho$ the bulk density of the surface layer material (kg m$^{-3}$), $C$ the specific heat of the surface layer material (J kg$^{-1}$K$^{-1}$), and $\lambda$ the thermal conductivity of the surface layer material (W m$^{-1}$ K$^{-1}$). $B(\mu)$ is recognized as the reciprocal Bowen ratio expressed in terms of a dimensionless parameter $\mu$, characterizing the relative role of surface temperature ($T_s$(K)) and surface (skin) specific humidity ($q_s$ (kg kg$^{-1}$)) on the surface energy budget. The "apparent thermal inertia of the air" $I_0$, characterizing the boundary-layer turbulence based on the Monin-Obukhov similarity theory, is given as follows [28].

$$I_0 = \rho_a C_p \sqrt{C_1 \kappa z}\left(C_2 \frac{\kappa z g}{\rho_a C_p T_0}\right)^{\frac{1}{6}},$$ (3)

where $\rho_a$ is the air density (kg m$^{-3}$), $\kappa$ the von Kármán constant ~0.4, $g$ the gravitational acceleration (m s$^{-2}$), $T_0$ the representative environment temperature (~300 K), $z$ the vertical distance (m) from the material surface above, $C_p$ the specific heat of air under constant pressure (103 J kg$^{-1}$ K$^{-1}$), $C_1$ and $C_2$ the parameters related to the universal constant in the empirical functions characterizing the atmospheric stability of the surface layer, $L_v$ the latent heat of vaporization of liquid water (2.5 × 10$^6$ J kg$^{-1}$), and $R_V$ the gas constant of water vapor (461 J kg$^{-1}$ K$^{-1}$).

The $q_s$ value in Equation (2) can be parameterized in terms of $T_s$ and $S_m$.

$$q_s = q_{sat}(T_s)\,\alpha(T_s, S_m),$$ (4)

$$q_{sat}(T_s) = \varepsilon\frac{e_0}{P_0}exp\left[\frac{L_v}{C_p R_v}\left(\frac{1}{T_0} - \frac{1}{T_s}\right)\right],$$ (5)

where $q_{sat}(T_s)$ is the saturation specific humidity at $T_s$, $\varepsilon$ the ratio of molecular weight of water vapor to dry air (0.622), $P_0$ the standard atmospheric pressure (~1000 hPa), $e_0$ the saturation vapor pressure of water at $T_0$ (11.6 hPa), and $\alpha$ the coefficient characterizing the water transport from inner soil pores to the soil surface. We used an alternative parameterization of $\alpha$ based on an analogy to surface $S_m$ dependence on the ratio of actual evapotranspiration to potential evapotranspiration. Thus,

$$\alpha = \left(\frac{S_m}{\varphi}\right)^\beta,$$ (6)

where $\varphi$ is the porosity of the soil (m$^3$ m$^{-3}$; $\varphi = 0.6$ at Ali site) and $\beta$ is a soil texture-dependent empirical parameter. Equation (6) represents $S_m$ determining the corresponding deduction of $q_s$ from

saturation condition $q_{sat}(T_s)$. Parameterization of $\alpha$ as in Equation (6) satisfies the physical constraints of $q_s$ (i.e., $q_s$ reaches its upper limit $q_{sat}(T_s)$ when soil is saturated ($S_m = \varphi$) and becomes zero when soil is completely dry) [29]. In this study, $\beta$ is estimated using Equation (6) by fitting the MEP modeled *LE* to the observed $LE_{EC}$ with the minimum root-mean-square error (RMSE).

## 3. Results

### 3.1. Estimation of LE by the MEP Model and Observational Data

The daily observed $S_m$ at the depth of 5 cm, $R_n$, and $T_s$ are applied in the estimation of *LE* (hereafter $LE_{MEP\,OBS}$) at the Ali site with the MEP model ($\beta = 1.1$) in summers from 2014 to 2016, in which $\beta$ is determined using Equation (6). The monthly values of $LE_{MEP\,OBS}$ and $LE_{EC}$ are given in Figure 1b. The coefficient of determination ($R^2$) and RMSE between $LE_{MEP\,OBS}$ and $LE_{EC}$ are 0.58 and 9.3 W m$^{-2}$ in summers, respectively (Figure 1c). From August 2014 to January 2017, the $R^2$ and RMSE between $LE_{MEP\,OBS}$ and $LE_{EC}$ are 0.64 and 6.6 W m$^{-2}$, respectively. Previous studies showed that the *LE* estimations with the RMSE values between 4.7 and 23.8 W m$^{-2}$ may be used to reveal the features of *LE* over the TP [3,13]. Thus, our resulting *LE* is acceptable, which implies that the MEP model can be used to estimate *LE* over the western TP.

### 3.2. LE Estimated by the MEP Model and Reanalysis Data and Its Long-Term Variation

Based on the daily $R_n$, $T_s$, and $S_m$ of the six reanalysis datasets in summers from 2014 to 2016, *LE* was estimated using the MEP model at the Ali site; hereafter referred to as the MEP and reanalysis (MR) *LE* and called $LE_{MEP\,ERA5}$, $LE_{MEP\,ERA\text{-}Interim}$, $LE_{MEP\,JRA\text{-}55}$, $LE_{MEP\,MERRA\text{-}2}$, $LE_{MEP\,NCEP\text{-}I}$, and $LE_{MEP\,NCEP\text{-}II}$, respectively. The $\beta$ value in the MEP model (Table 2) is calibrated using Equation (6) by fitting the MEP-estimated *LE* to $LE_{EC}$ for each reanalysis dataset. Figure 2 shows the monthly MR *LE*. The monthly average value of $LE_{EC}$ is 11.1 W m$^{-2}$ in summers from 2014 to 2016. For MR *LE*, the monthly average values of $LE_{MEP\,ERA5}$, $LE_{MEP\,ERA\text{-}Interim}$, $LE_{MEP\,JRA\text{-}55}$, $LE_{MEP\,MERRA\text{-}2}$, $LE_{MEP\,NCEP\text{-}I}$, and $LE_{MEP\,NCEP\text{-}II}$ are 9.2, 14.2, 16.5, 10.8, 12.0, and 8.7 W m$^{-2}$, respectively. The $R^2$ values of $LE_{MEP\,ERA5}$, $LE_{MEP\,JRA\text{-}55}$, and $LE_{MEP\,MERRA\text{-}2}$ are 0.43, 0.34, and 0.54, respectively (Table 3). These high correlations imply the consistency between the three MR *LE* and $LE_{EC}$, especially for $LE_{MEP\,MERRA\text{-}2}$. The $R^2$ values of $LE_{MEP\,ERA\text{-}Interim}$, $LE_{MEP\,NCEP\text{-}I}$, and $LE_{MEP\,NCEP\text{-}II}$ are between 0.11 and 0.15, which show relatively poor correlations between these three MR *LE* and $LE_{EC}$. It is evident that the $LE_{MEP\,MERRA\text{-}2}$ RMSE (10.1 W m$^{-2}$) and mean bias error (MBE) ($-0.34$ W m$^{-2}$) are smallest among the six MR *LE* values.

**Table 2.** The values of $\beta$ in the maximum entropy production model for six reanalysis datasets at the Ali site in summers from 2014 to 2016.

|   | ERA5 | ERA-Interim | JRA-55 | MERRA-2 | NCEP-I | NCEP-II |
|---|------|-------------|--------|---------|--------|---------|
| $\beta$ | 1.1 | 2.4 | 4.0 | 8.2 | 4.1 | 3.5 |

Based on the above statistical indices, $LE_{MEP\,ERA5}$, $LE_{MEP\,JRA\text{-}55}$, and $LE_{MEP\,MERRA\text{-}2}$ perform better than the other three MR *LE*. Xie and Wang (2019) suggested that merging data through various data sources can reduce the uncertainties of data. Therefore, to achieve a better *LE* estimation, we used the averages of daily $R_n$, $T_s$, and $S_m$ of ERA5, JRA-55, and MERRA-2 as a new merged dataset for re-estimating the *LE* with the MEP model (hereafter referred to as *LE*-merged). Here, the value of $\beta$ is equal to 3.4, which is calibrated using Equation (6) by fitting *LE*-merged to $LE_{EC}$. The $R^2$, RMSE, and MBE values of *LE*-merged are 0.57, 9.3 W m$^{-2}$, and $-2.25$ W m$^{-2}$, respectively. This result indicates that *LE*-merged is a better estimation compared with the *LE* estimation using the single reanalysis data source and the MEP model.

**Table 3.** The coefficient of determination ($R^2$), the root-mean-square error (RMSE), and the mean bias error (MBE) of latent heat flux (*LE*) estimated by reanalysis datasets (reanalysis *LE*) and *LE* estimated from the maximum entropy production model and the reanalysis datasets (MR *LE*) at the Ali site in summers from 2014 to 2016.

|  | LE | $R^2$ | RMSE (W m$^{-2}$) | MBE (W m$^{-2}$) |
|---|---|---|---|---|
| **ERA5** | **Reanalysis** | 0.07 | 46.8 | 31.0 |
|  | **MR** | 0.43 | 11.2 | −3.5 |
| **ERA-Interim** | **Reanalysis** | 0.14 | 33.7 | 28.8 |
|  | **MR** | 0.15 | 12.8 | −0.9 |
| **JRA-55** | **Reanalysis** | 0.13 | 42.3 | 36.1 |
|  | **MR** | 0.34 | 19.1 | 5.6 |
| **MERRA-2** | **Reanalysis** | 0.38 | 15.8 | 8.4 |
|  | **MR** | 0.54 | 10.1 | −0.34 |
| **NCEP-I** | **Reanalysis** | 0.01 | 52.1 | 49.3 |
|  | **MR** | 0.14 | 12.8 | 0.10 |
| **NCEP-II** | **Reanalysis** | 0.09 | 48.4 | 44.5 |
|  | **MR** | 0.11 | 13.2 | 0.20 |

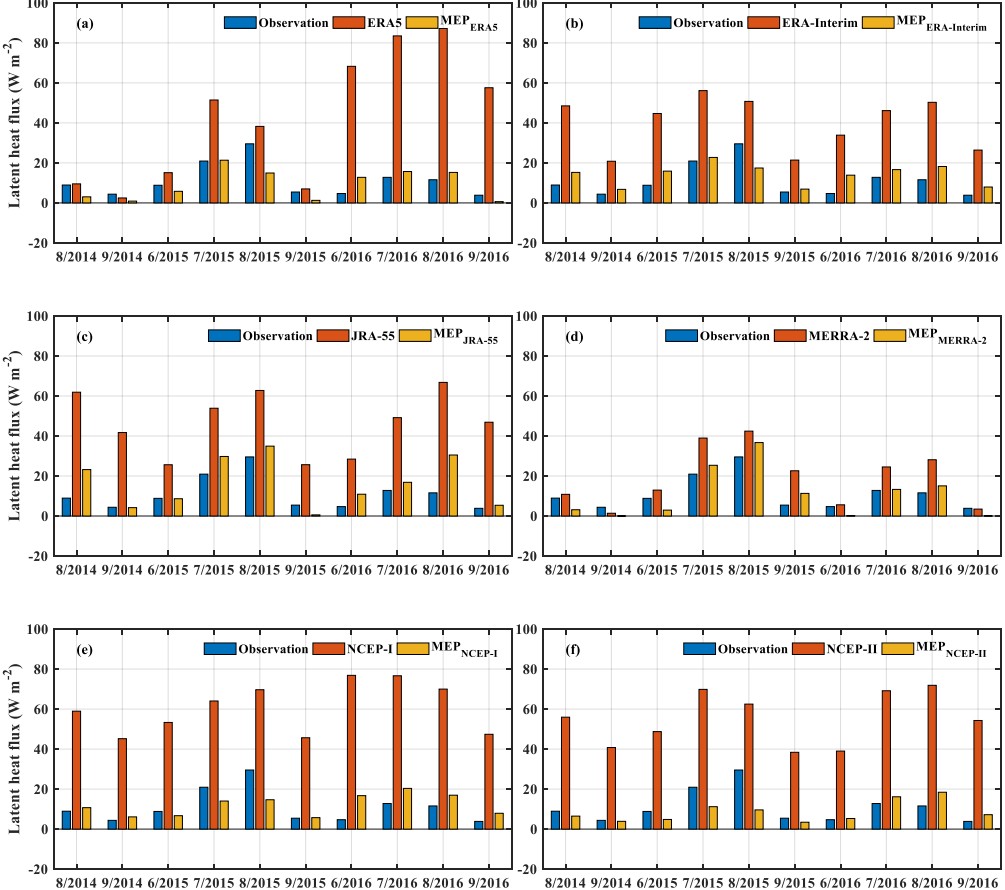

**Figure 2.** The summer monthly values of the observed latent heat flux ($LE_{EC}$), the latent heat flux (*LE*) of the reanalysis datasets (reanalysis *LE*), and *LE* estimated by the maximum entropy production (MEP) model and the reanalysis datasets (MR *LE*) at the Ali site from 2014 to 2016 ((**a**) ERA5, (**b**) ERA-Interim, (**c**) JRA-55, (**d**) MERRA-2, (**e**) NCEP-I, and (**f**) NCEP-II).

The MR *LE* is further compared with *LE* provided by the six reanalysis datasets (hereafter referred to as $LE_{ERA5}$, $LE_{ERA-Interim}$, $LE_{JRA-55}$, $LE_{MERRA-2}$, $LE_{NCEP-I}$, and $LE_{NCEP-II}$, respectively). The monthly values of $LE_{ERA5}$, $LE_{ERA-Interim}$, $LE_{JRA-55}$, $LE_{MERRA-2}$, $LE_{NCEP-I}$, and $LE_{NCEP-II}$ are 42.0, 39.9, 46.3, 19.1, 60.8, and 55.1 W m$^{-2}$, respectively, and are significantly greater than those of the MR *LE* and $LE_{EC}$

(Figure 2). Clearly, the RMSE values of the six MR *LE* are reduced (Table 3). *LE*-merged performs best compared with the MR *LE* and the reanalysis *LE*. Hence, $R_n$, $T_s$, and $S_m$ of ERA5, JRA-55, and MERRA-2 can be applied in driving the MEP model to estimate *LE* at the Ali site over the western TP.

To explore the variability of *LE* at the Ali site, the long-term series of *LE*-merged is analyzed in this study. In Figure 3, *LE*-merged generally fluctuates between 1.8 and 18.1 W m$^{-2}$, with the mean value of 7.4 W m$^{-2}$, and showing a decreasing trend of $-7.4$ W m$^{-2}$ decade$^{-1}$ during 1980–1991 and an increasing trend of 1.8 W m$^{-2}$ decade$^{-1}$ during 1992–2018. For the whole period 1980–2018, *LE*-merged generally presents an increasing trend of 0.3 W m$^{-2}$ decade$^{-1}$. The MR *LE* and the reanalysis *LE* of ERA5, JRA-55, MERRA-2, and NCEP-I show the similar variations to *LE*-merged, with the decreasing trend during 1980–1991 and the increasing trend during 1992–2018 (Figure 4a,c–e), especially for both $LE_{\text{MEP MERRA-2}}$ and $LE_{\text{MERRA-2}}$ (Figure 4d). But the features of the ERA-Interim and NCEP-II MR *LE* and reanalysis *LE* series are quite different from those of *LE*-merged (Figure 4b,f).

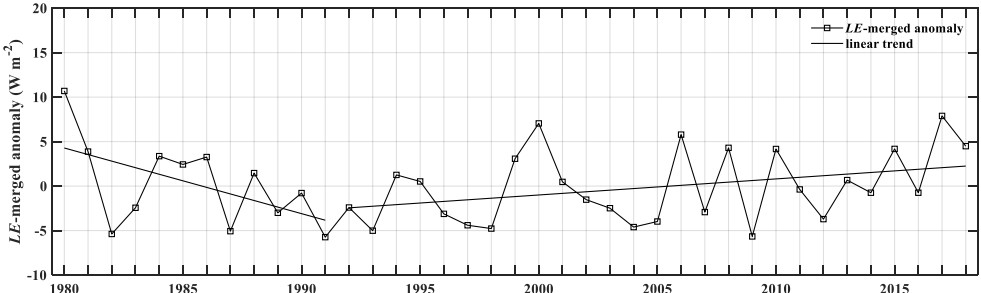

**Figure 3.** The temporal series of the summer latent heat flux anomaly estimated by the maximum entropy production model and the merged dataset (*LE*-merged) at the Ali site from 1980 to 2018 and its linear trends.

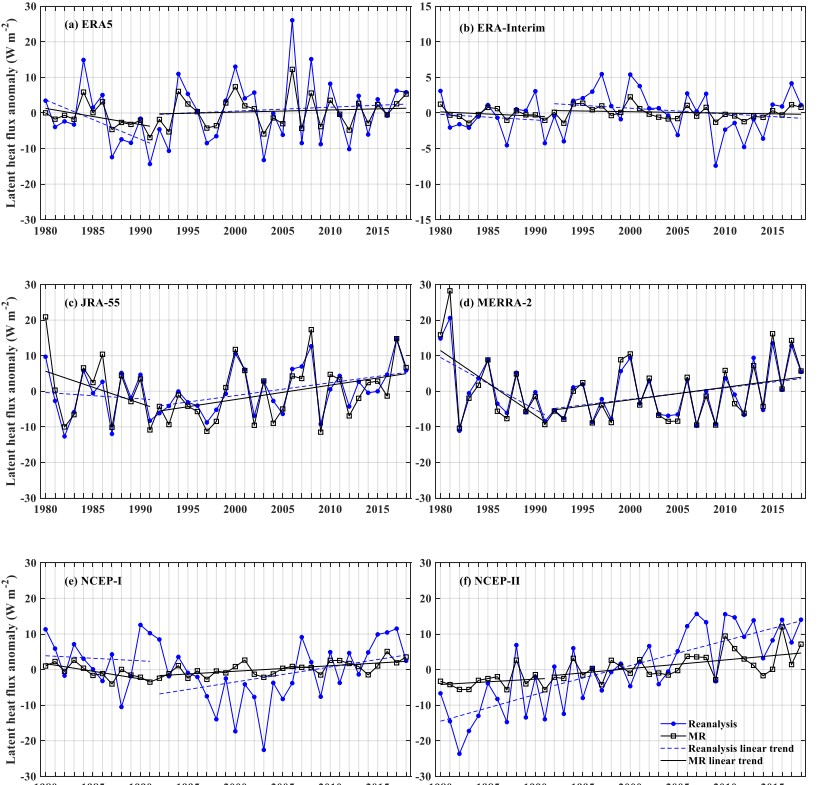

**Figure 4.** The temporal series of the summer reanalysis *LE* and MR *LE* anomalies at the Ali site from 1980 to 2018 and their linear trends.

## 4. Discussion

Previous studies indicated that surface energy fluxes during summer are mainly affected by precipitation, especially in arid regions [30–32]. To explore the reasons for the difference in *LE* of the six reanalysis datasets, we analyzed the summer reanalysis precipitation. The varying features of the summer reanalysis precipitation are generally similar to that of the observed precipitation during 2014–2016, with the maximum precipitation occurring in July-August (Figure 5). However, the summer-averaged precipitation values of the six reanalysis datasets are generally greater than the observation (0.64 mm), especially for JRA-55 (1.7 mm), NCEP-I (5.6 mm), and NCEP-II (7.1 mm). The summer-averaged precipitation values of ERA5, ERA-Interim, and MERRA-2 are 1.0, 0.83, and 0.65 mm, respectively. The RMSE values of the precipitation of ERA5, ERA-Interim, JRA-55, MERRA-2, NCEP-I, and NCEP-II are 2.4, 2.3, 2.6, 2.2, 5.9, and 7.4 mm, respectively. The RMSE of MERRA-2 precipitation is smallest. Based on the above analysis, the MERRA-2 *LE* and precipitation perform best among the six reanalysis datasets. Therefore, the errors in precipitation may lead to a large uncertainty in the reanalysis *LE*. In addition, the difference in the parameterization of the moisture transfer coefficient, surface roughness lengths, and wind speed may also lead to the conspicuous differences among the six reanalysis *LE* data [33–35].Since the parameterization scheme of the MEP model may be able to reflect the surface humidity condition, the uncertainties of the estimated *LE* using the MEP model can be substantially reduced compared with the reanalysis *LE* [20,29].

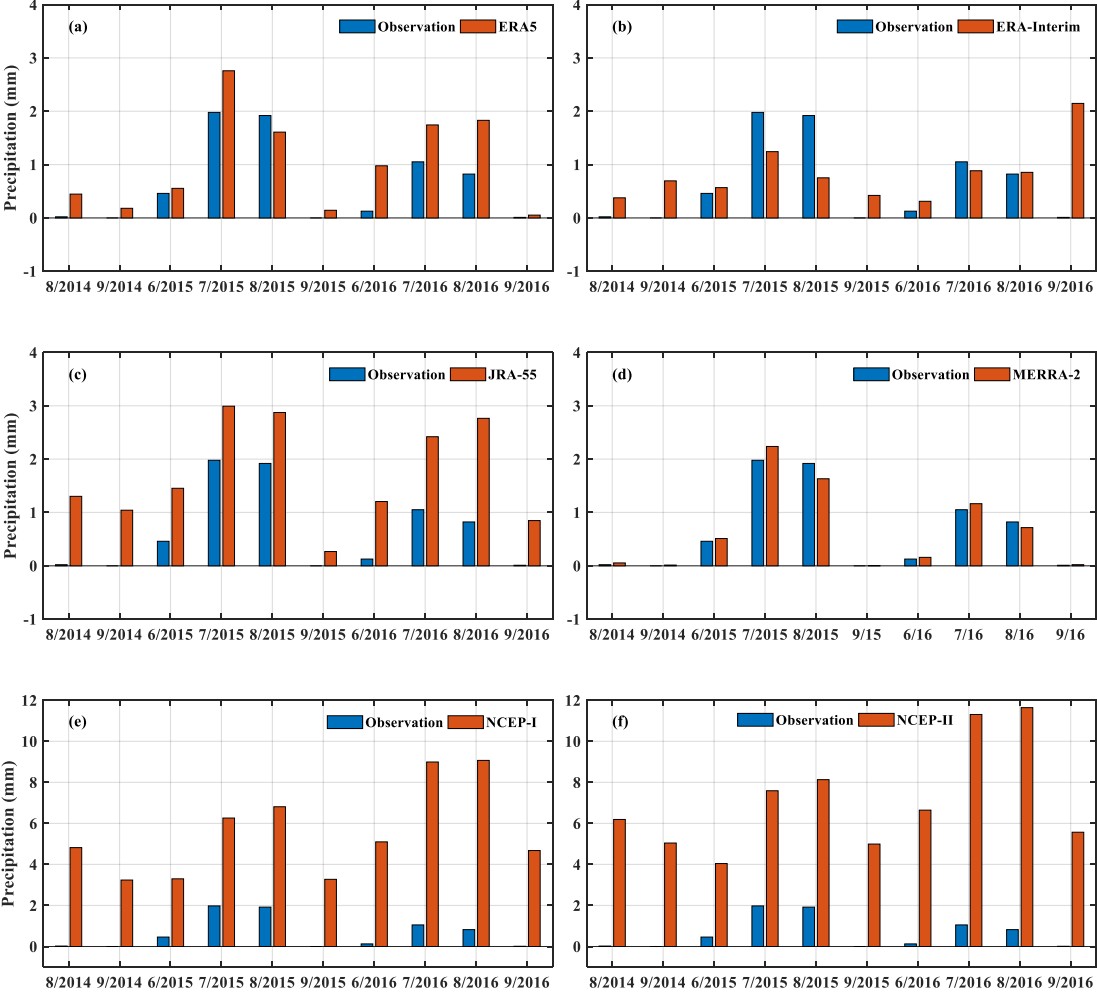

**Figure 5.** The observed and reanalysis monthly precipitation values in summers at the Ali site from 2014 to 2016.

The *LE*-merged presents a significant variation from 1980 to 2018. Han et al. (2017) reported that *LE* presented an upward trend between 0 and 2 W m$^{-2}$ year$^{-1}$ from 2001 to 2012. During this period, *LE*-merged also shows an upward trend of 0.8 W m$^{-2}$ decade$^{-1}$. It is evident that *LE* over the western TP shows a complicated feature during a long period under the influence of climatic and environmental factors.

Previous studies showed that the increased $S_m$ could lead to an increase of *LE* [36–38]. The increase (decrease) of $S_m$ may bring high (low) potential evaporation, leading to the increase (decrease) in *LE*. Figure 6a shows the time series of summer $S_m$ averaged over the ERA5, JRA-55, and MERRA-2 reanalysis datasets at the Ali site during 1980–2018. It is seen that summer $S_m$ at the Ali site presents an overall increasing trend of 0.06 decade$^{-1}$ during 1980–2018, which may contribute to the increase of *LE*-merged during this period. Moreover, $S_m$ also presents a decreasing trend of −0.082 decade$^{-1}$ from 1980 to 1991 (corresponding to the decreasing trend of *LE*-merged) and an increasing trend of 0.022 decade$^{-1}$ from 1992 to the 2018 (corresponding to the increasing trend of *LE*-merged). The $R^2$ value between $S_m$ and *LE*-merged is 0.96. The minimum (maximum) values of both *LE*-merged and $S_m$ occur in 1991 (1980). Therefore, the high correlation between $S_m$ and *LE*-merged implies that the long-term variation of *LE*-merged may be mainly controlled by the local $S_m$.

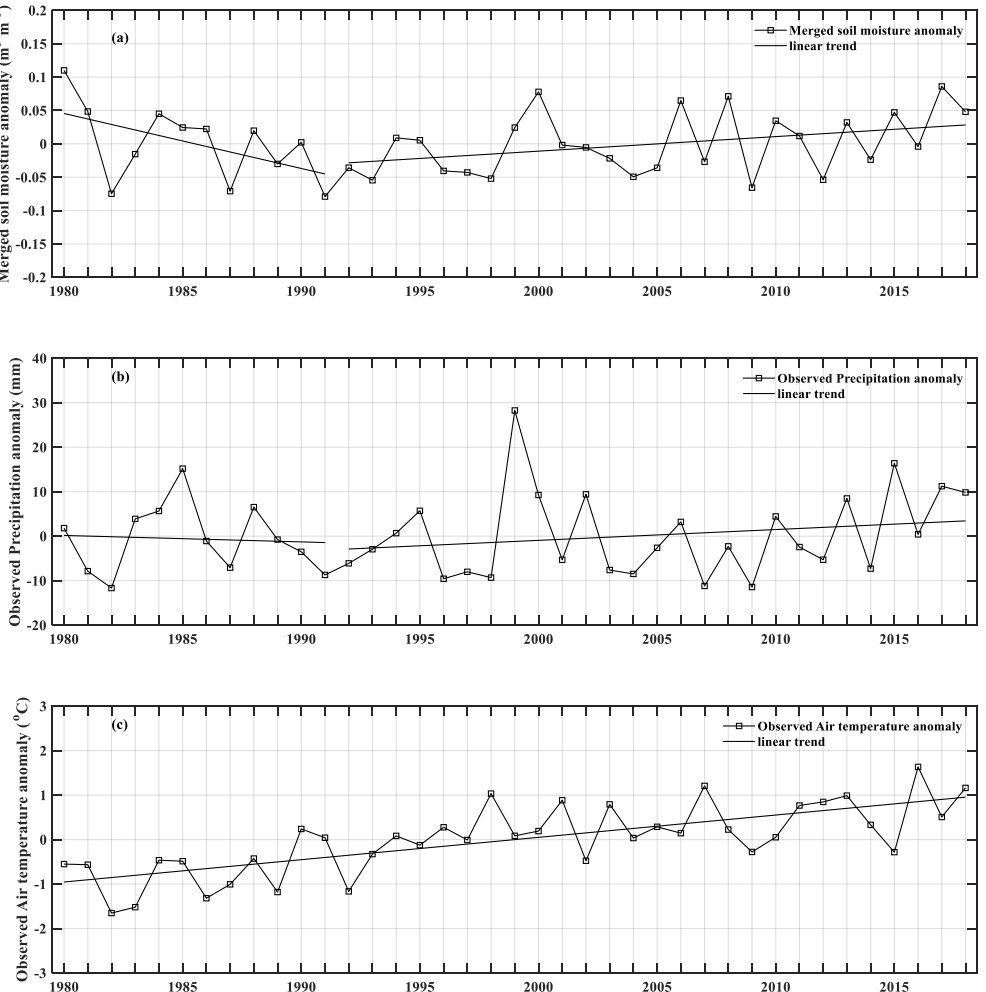

**Figure 6.** The temporal series of (**a**) the summer soil moisture ($S_m$) anomaly of the merged dataset at the Ali site during 1980–2018; (**b**) same as in (**a**) but for the summer observed precipitation; (**c**) same as in (**a**) but for the summer observed air temperature.

The variation of $S_m$ is dramatically influenced by groundwater cycles such as precipitation, lake, and glacier [39]. Our result shows that the summer precipitation at the Ali site presents a

decreasing trend of $-1.5$ mm decade$^{-1}$ during 1980–1991 and an upward trend of 2.4 mm decade$^{-1}$ during 1992–2018, with an overall increasing trend of 1.1 mm decade$^{-1}$ during 1980–2018 (Figure 6b). This variation of precipitation is consistent with that of $S_m$. The summer $T_a$ presents an upward trend of 0.5 °C decade$^{-1}$ during 1980–2018 (Figure 6c). In addition to the decrease of precipitation, the lake area had also decreased near the Ali site from the early 1980s to the early 1990s [40]. This may be mainly caused by the combination effects of $T_a$, precipitation, and groundwater cycles during this period. Afterwards, under the background of the warming and wetting climate of TP, the retreat amplitudes of most of glaciers over the TP and surrounding regions appear to be accelerating and reach a maximum in the 1990s [41,42]. Yao et al. (2007) reported that the magnitude of glacial retreat is largest in the Karakorum Mountains (near the Ali site) in the 1990s. At the same time, the runoff of some rivers in this region had also increased due to the glacial melting. Zhang et al. (2019) further showed that the total lake area had been increasing near the Ali site since the mid-1990s as a response to a hydrological cycle intensified by climate changes. Although $T_a$ and *LE*-merged have been increasing during this period (which may cause the decrease in lake area), the effects of the increasing precipitation and accelerating glacial retreat possibly still dominate the expansion of the local lake area. Therefore, the increasing precipitation and accelerating glacial retreat may result in the increase of $S_m$ at the Ali site after the 1990s. On the whole, the increase of $S_m$ at the Ali site during 1980–2018 may be mainly caused by the increase of summer precipitation.

Moreover, a warmer and wetter environment may help vegetation growth. Figure 7 shows the summer NDVI values during 1981–2018, in which the NDVI values during 1981–2015 are obtained from GIMMS3g and those during 2000–2018 are obtained from MODIS. Both of GIMMS3g and MODIS NDVI show increasing trends during 2000–2015. GIMMS3g NDVI shows a decreasing (an increasing) trend during 1981–1991 (1992–2015). It is evident that corresponding to the increasing trend of $T_a$ and the decreasing trend of precipitation during 1981–1991, NDVI presents a decreasing trend. While corresponding to the increasing trends of both $T_a$ and precipitation during 1992–2018, NDVI shows a general increasing trend. NDVI may affect variations of *LE* through modulating surface vegetation evapotranspiration and surface roughness, and is positively correlated with the land surface energy and water cycles [43,44]. Therefore, the consistent trends of *LE*-merged and NDVI (shown in Figures 3 and 7) suggest a possible effect of NDVI on *LE* at the Ali site.

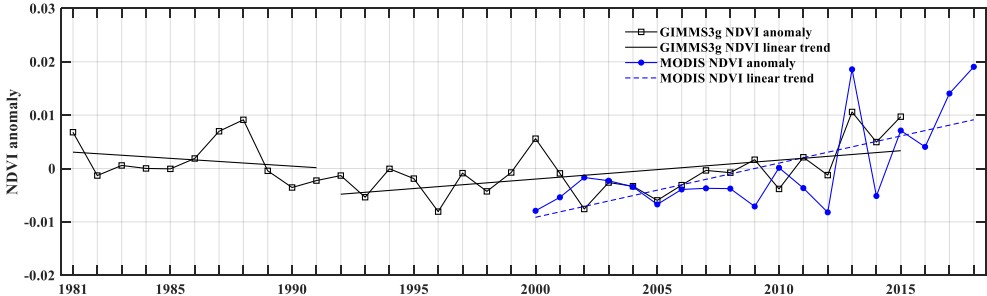

**Figure 7.** The temporal series of the summer normalized difference vegetation index (NDVI) at the Ali site during 1981–2018 and its linear trends.

## 5. Conclusions

In this study, we applied the MEP model and the ensemble average of $R_n$, $T_s$, and $S_m$ of the ERA5, JRA-55, and MERRA-2 reanalysis datasets to re-estimate summer *LE* at the Ali site over the western TP. During 1980–2018, the variation of *LE* is consistent with that of $S_m$. The latter is influenced by climatic and environmental factors, such as precipitation, glaciers, and near-surface water conditions. *LE* generally shows a consistent trend with NDVI. Our study not only tends to agree with the increasing trends of *LE* in previous studies at the Ali site, but also reveals the long-term variation of *LE* and its relationship to the climatic and environmental factors. However, we explore only one site of the western TP in this study. The *LE*-merged should be further extended to more sites for a longer period,

in combination with other surface heat sources, to investigate the intraseasonal variability of *LE* [45] and its association with the climate and environment in the future studies.

**Author Contributions:** Methodology, J.W. and Y.D.; supervision, P.Z., J.W. and Y.D.; writing—original draft, N.L.; writing—review and editing, P.Z. All authors have read and agreed to the published version of the manuscript.

**Funding:** This research was funded by the National Key Research and Development Plan Project (2018YFC1505700) and NSFC (91637312).

**Acknowledgments:** This work is jointly sponsored by the National Key research and Development Plan Project (2018YFC1505700) and NSFC (91637312). The TIPEX-III datasets come from the website (http://data.cma.cn/tipex). The normalized difference vegetation index of the MODIS/Terra Vegetation Indices can be obtained from the website (https://e4ftl01.cr.usgs.gov/MOLT/MOD13Q1.006/). The normalized difference vegetation index of Advanced Very High Resolution Radiometer-Based Global Inventory Modelling and Mapping Studies can be downloaded from the website (https://ecocast.arc.nasa.gov/data/pub/gimms/3g.v1/). The ERA5, ERA-Interim, JRA-55, MERRA-2, NCEP-I, and NCEP-II reanalysis datasets come from their websites.

**Conflicts of Interest:** The authors declare no conflict of interest.

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
