# Peer review of "The Long-Term Change of Latent Heat Flux over the Western Tibetan Plateau"

_atmosphere, doi:10.3390/atmos11030262_

Round 1

Reviewer 1 Report

This study estimated the summer latent heat flux (LE) using the maximum entropy production model driven by the net radiation, surface temperature, and soil moisture from three reanalysis datasets (ERA5, JRA-55, and MERRA-2) at the Ali site over the western TP during 1980–2018.

In general, the manuscript is well written, the logic is clear and the evidences can support their conclusions. Therefore, I think this manuscript can be published in Atmosphere after a minor revision.

Below are the reviewer’s suggestions/concerns for the authors’ consideration.

Although the coordinate has been given in the text, it is better to show the specific location of Ali site in the map as added in Figure 1.

Figure 7, I am confused by the blue line in this figure. Is it a 9-year moving average or linear trend?

Given that the daily data are available, the author may want to extend their future study to intraseasonal timescale, the below relevant paper is for the authors reference.

Li W., et al. 2020: Intraseasonal Variability of Tibetan Plateau Snow Cover. Int. J. Climato., doi: 10.1002/joc.6407.

Reviewer 2 Report

Comments and suggestions were included to the document 

Reviewer 3 Report

The title is not proper, as the authors' conclusions are based on rainfall and soil moisture and not temperature.

Line 78. The link to the website does not work.

Line 90. It is not clear the monthly observed LE values presented are monthly averages, maximums or minimums.

Line 104-105. The link to the website does not work.

Line 109. It is very hard to know in detail the meaning of the parameters of equations 1 and 2. It is distressing to have to refer to another paper to understand the meaning of the parameters used in this manuscript.

Line 113. The estimation of the specific saturation humidity at ?? according to the Clausius-Clapeyron, is not very clear. It should be developed more clearly. See equation (1) in:

Agard, V., & Emanuel, K. (2017). Clausius–Clapeyron Scaling of Peak CAPE in Continental Convective Storm Environments. Journal of the Atmospheric Sciences, 74(9), 3043-3054. doi: 10.1175/jas-d-16-0352.1

Line 118-120. Once again, another paper is used to prove the parameters employed in equation 4. This strongly queries the scientific contribution of this manuscript.

Line 121. The results should be separate from the discussion. The reading of the manuscript is very confused in this way.

Line 124. After reading the paper, I don’t know how you calculate/get the beta coefficient. 

Line 132. In Figure 1b it is not clear what the meaning of the equation x,y

Line 141-142 and 170. It is necessary to show how the beta calibration procedure is performed.

Line 179. From this line, the authors use precipitation as a variable to prove the variations of the analyzed series. At this point, it is not known if it is part of the results or of the discussion. 

Line 201. One more evidence of the confusing aspect of the manuscript is that conclusions are presented within the results section.

Line 208. In figure 4c it is not clear which units the abscissa axis has.

Line 210. What is the reason for showing the historical series (figure 4), in the results section? should be presented earlier in the manuscript. In addition, once again x,y equations are presented without any sense.

Line 217. The abscissa axis in figure 5b has no meaning.

Line 224-230. These conclusions cannot be reached from figures 5. At least one autocovariance or serial correlation function is required to know the homogeneous ten-year periods.

Line 243. The figures 6, are confusing, what do the letters X and Y mean? How can an equation be obtained when the X-axis is the years?

Line 252-253. No graphs or historical data on precipitation during these years are presented to verify this statement.

Line 262. Figure 7b again presents equations with X,Y that makes no sense in the context of the manuscript.

Line 285-286. The results presented in the manuscript cannot support that the NDVI factor affects the variation of LE and that it is related to evapotranspiration.

General comments

In general, the manuscript does not allow us to understand how the Maxima Entropy procedure is used.

The conclusions section does not present any conclusions. The conclusions section has discussions of the results. The conclusions can be read in the results section.

It is an article oriented exclusively to the application of existing models and databases. It is not clear what the scientific contribution of the work is.

Round 2

Reviewer 2 Report

Comments about “The long-term change of latent heat flux over the western Tibetan Plateau”

General comments

The article estimates the summer latent heat flux (LE) at the Ali site over the western Tibetan Plateau (TP) and analyzes its changes during 1980–2018 using three reanalysis datasets (ERA5, JRA-55, and MERRA-2). The title in the new version is more concise and I agree with this change. All my comments and suggestions indicated in the document were well answered and explained fully.

Abstract

The authors completed the abstract as was required.

Introduction

Here again the authors adjusted the text of the document as requested.

Data and methods

Figure 1a was added to show the geographic location of the analyzed site and Table 1 was completed with the SHEC data as requested. In this way, the minor importance of LE as a dissipation process in that place is clearly stated. Also, the equations now are more detailed and additional information was aggregate to complete de methodology.

Results Discussion

The writing of results was thoroughly reviewed. The new version was rearranged and the confused information (Figures) was not included finally.

Conclusions

The conclusions set out in a more concrete way the main contributions of the research.

Reviewer 3 Report

Accept in present form